# Benefit of Atrial Overdrive Pacing in Patients with Sleep Apnea: A Meta-Analysis

**DOI:** 10.3390/jcm10184065

**Published:** 2021-09-09

**Authors:** Nithi Tokavanich, Pattranee Leelapatana, Somchai Prechawat, Voravut Rungpradubvong, Wimwipa Mongkonsritrakoon, Saraschandra Vallabhajosyula, Narut Prasitlumkum, Charat Thongprayoon, Wisit Cheungpasitporn, Ronpichai Chokesuwattanaskul

**Affiliations:** 1Division of Cardiology, Department of Medicine, Faculty of Medicine, Chulalongkorn University and King Chulalongkorn Memorial Hospital, Thai Red Cross Society, Bangkok 10330, Thailand; tuck.pattranee@gmail.com (P.L.); s_prechawat@hotmail.com (S.P.); vutvut33@hotmail.com (V.R.); 2Department of Pediatric, Chulalongkorn University and King Chulalongkorn Memorial Hospital, Thai Red Cross Society, Bangkok 10330, Thailand; mwimwipa@gmail.com; 3Section of Cardiovascular Medicine, Department of Medicine, Wake Forest University School of Medicine, Winston-Salem, NC 27101, USA; svallabh@wakehealth.edu; 4Division of Cardiology, University of California Riverside, Riverside, CA 92521, USA; narutprasitlumkum@gmail.com; 5Department of Medicine, Mayo Clinic, Rochester, MN 55905, USA; charat.thongprayoon@gmail.com (C.T.); wcheungpasitporn@gmail.com (W.C.); 6Center of Excellence in Arrhythmia Research Chulalongkorn University, Department of Medicine, Faculty of Medicine, Chulalongkorn University, Bangkok 10330, Thailand

**Keywords:** atrial overdrive pacing, cardiac implantable electronic device, central sleep apnea, obstructive sleep apnea, sleep apnea, systematic reviews, meta-analysis

## Abstract

Background: Sleep apnea is one of the most common conditions around the world. This disorder can significantly impact cardiovascular morbidity and mortality. Atrial overdrive pacing (AOP) is a treatment modality that can potentially decrease respiratory events. There is currently a lack of evidence to confirm the benefits of AOP. We aimed to assess the impact of AOP in patients with obstructive sleep apnea (OSA), central sleep apnea (CSA), and mixed type. Methods: A literature search for studies that reported the impact on apnea–hypopnea index (AHI) by cardiac implantable electronic devices with different pacing modes was conducted using MEDLINE, Embase, and Cochrane Database from inception through July 2020. Pooled standard mean difference with 95%CI was calculated using a random-effects model. Results: Fifteen studies, including thirteen randomized studies and two observational studies containing 440 patients, were identified. The standard mean difference in apnea–hypopnea index of atrial overdrive pacing demonstrated less duration of apnea/hypopnea in patients with atrial overdrive pacing (AOP) (SMD −0.29, 95%CI: −0.48, −0.10, I^2^ = 57%). Additional analysis was performed to assess the effect of atrial overdrive pacing in patients with or without severe sleep apnea syndrome (mean AHI < 30 defined as non-severe). There was no statistically significant difference in standardized mean in AHI in both subgroups between AOP and control groups (SMD −0.25, severe sleep apnea syndrome SMD −0.03, I^2^ = 0.00%). Conclusions: AOP was associated with a statistically significant reduction in AHI, but the magnitude of reduction was small. AOP may potentially be used as an adjunctive treatment with other modalities in treating patients with sleep apnea.

## 1. Introduction

Sleep apnea is an important risk factor for many diseases. Sleep apnea is divided into three subtypes: obstructive sleep apnea (OSA), central sleep apnea (CSA), and mixed sleep apnea [1]. This disorder is strongly associated with many cardiovascular diseases, including coronary artery disease, hypertension, and cardiac arrhythmia, causing significant morbidity and mortality. Its impact can be explained by enhancing the platelet function, which causes systemic inflammation due to chronic hypoxemia and chronic hypercarbia. There are other mechanisms involved in the development of cardiovascular complications in patients with sleep apnea, such as the interplay between sympathetic stimulation, alteration of vascular regulating system, endothelial dysfunction, and oxidative stress [2]. Several studies revealed that untreated sleep apnea patients had higher all-cause mortality compared with healthy individuals [3,4]. There is also an association between sleep apnea and psychological problems [5]. The prevalence of sleep apnea around the globe is 22% in men and 17% in women. The total number of individuals affected globally by sleep apnea has been estimated to be between 711 and 961 million, and the figure is now on the rise [6,7,8]. Early diagnosis and treatment of patients with sleep apnea can produce significant benefits both medically and economically [9,10].

Atrial overdrive pacing (AOP) is a pacemaker mode designed to increase the atrial pacing rate to a level slightly higher than a patient’s intrinsic rate. This mode of pacing was initially designed to suppress premature atrial contraction (PAC). This type of pacing should therefore decrease the incidence of atrial tachyarrhythmia [11]. There are many theories explaining why AOP can improve sleep apnea. One of these theories is that increasing cardiac output from AOP leads to a reduction in lung circulation time and left ventricular filing time, which can stabilize breathing [12,13]. Currently, insufficient data exist to confirm any potential benefit of pacing on sleep apnea. To fill this research gap, we conducted a meta-analysis and systematic review to assess the potential usefulness of AOP in patients with sleep apnea.

## 2. Methods

### 2.1. Literature Review and Search Strategy

The protocol for this meta-analysis is registered with PROSPERO (International Prospective Register of Systematic Reviews; no. CRD 42020203899). A systematic literature search of MEDLINE (1946 to July 2020), Embase (1988 to July 2020), and the Cochrane Database of Systematic Reviews (database inception to July 2020) was conducted to assess the impact of AOP in patients with sleep apnea.

The systematic literature review was undertaken independently by two investigators (R.C. and N.T.) applying a search approach that incorporated the term “atrial overdrive pacing” and “sleep apnea”, which is provided in online Appendix A. A manual search for conceivably relevant studies using references of the included articles was also performed. No language limitation was applied. This study was guided by the STROBE (Strengthening the Reporting of Observational Studies in Epidemiology) and the Preferred Reporting Items for Systematic Reviews and Meta-Analysis (PRISMA) statement [14].

### 2.2. Selection Criteria

Eligible studies were limited to observational (cohort, case-control, or cross-sectional studies) and randomized studies that reported a sleep apnea-related clinical index in AOP patients. Studies must have provided data on the clinical characteristics, sleep apnea-related index, and cardiac implantable electronic device types. Inclusion was not limited by study size. Case reports were excluded. Retrieved articles were individually reviewed for their eligibility by the two investigators (R.C. and N.T.). Discrepancies were discussed and resolved by a third researcher (S.P.). The Newcastle–Ottawa quality assessment scale was used to determine the quality of study for case-control studies and outcomes of interest for cohort studies [15]. A modified Newcastle–Ottawa scale was used for cross-sectional studies [16]. The risk of bias by the Cochrane Collaboration tool was used for assessing the risk of bias in randomized trials.

### 2.3. Data Abstraction

A structured data collection form was utilized to collect the following information from each study, including title, study year, first author name, publication year, study country, demographic characteristics, and type of AOP device.

### 2.4. Statistical Analysis

Analyses were performed using R software version 3.6.3 (R Foundation for Statistical Computing, Vienna, Austria). The raw data for this systematic review are publicly available through the Open Science Framework (URL: osf.io/3f2pb, accessed on 27 August 2021). Adjusted point estimates from each included study were combined using the generic inverse variance approach of DerSimonian and Laird, which designated the weight of each study based on its variance [17]. Given the possibility of between-study variance, we used a random-effects model rather than a fixed-effects model due to anticipated heterogeneity across studies. Cochran’s Q test and I^2^ statistics were applied to determine the between-study heterogeneity. A value of I^2^ of 0–25% indicates insignificant heterogeneity, 26–50% low heterogeneity, 51–75% moderate heterogeneity, and 76–100% high heterogeneity [18]. Publication bias was assessed via the Egger test [19].

## 3. Results

A total of 1465 potentially eligible articles were identified using our search strategy. After the exclusion of 1442 duplicate articles, case reports, correspondences, review articles, in vitro studies, pediatric patient population, or animal studies, 23 articles remained for full-length review. Eight were excluded as the outcomes of interest were not reported.

The final analysis included a total of 15 studies (13 randomized studies and 2 observational studies [13,20,21,22,23,24,25,26,27,28,29,30,31,32,33] containing a total of 440 patients. The literature retrieval, review, and selection process are demonstrated in Figure 1. The characteristics and quality assessment of the included studies are presented in Table 1 and Figure 2.

### 3.1. Apnea–Hypopnea Index (AHI) in AOP Patients

The standardized mean difference (SMD) in the apnea–hypopnea index (AHI) demonstrated significantly less duration of apnea/hypopnea in patients with AOP, SMD −0.29 (95%CI: −0.48 to −0.10, I^2^ = 57%, Figure 3).

Furthermore, when the meta-analysis was conducted to assess the effect of AOP in patients with or without severe sleep apnea syndrome (mean AHI < 30 defined as non-severe) [34], there were no statistically significant differences in the standardized mean of AHI in both subgroups between AOP and control groups (non-severe sleep apnea syndrome SMD −0.25 (95%CI: −0.74 to 0.23, I^2^ = 0.00%), severe sleep apnea syndrome SMD −0.03 (95%CI: −0.30 to 0.23, I^2^ = 0.00%), Figure 4). Meta-regression analyses showed no significant correlations between the study year and the standardized mean difference of AHI in patients (*p* = 0.13).

### 3.2. Evaluation for Publication Bias

Funnel plots (Figure 5) and Egger’s regression asymmetry tests were performed to assess for publication bias among selected studies. No significant publication bias was found (*p* = 0.54).

## 4. Discussion

Although AOP is not widely used as a treatment for patients with sleep apnea, our systematic review demonstrated the benefit of this method. AOP has shown a benefit for patients suffering from sleep apnea, with a mean difference of around −0.29 on the apnea–hypopnea index (AHI). Similar results were found in a previous meta-analysis and systematic review completed by Baranchuk et al. [35] in which AOP delivered benefits to this group of patients. The difference between our study and the previous meta-analysis is that the latter only included 10 studies and 175 patients, and our meta-analysis included 440 patients, and patients were not classified according to sleep apnea severity. Our study also found that the benefit of AOP was not apparent in both groups. This finding implies that the pathophysiology of sleep apnea was explained by a complex interplay between the respiratory and cardiovascular systems. Despite the positive results of AOP in sleep apnea patients, it was not the only therapy given to patients in our analysis. Other therapies included those proven to have clear benefits, such as continuous positive pressure (CPAP) and treatment of heart failure [20,21,25,27,28,30].

The most effective method in treating patients with sleep apnea, especially OSA type, is CPAP, which is considered the gold standard. The AHI in OSA patients between pre and post treatment of CPAP was reduced to around 33.3–48.3% [36]. The network meta-analysis showed a reduction in AHI with CPAP by 25.27 events/hours [37]. Other methods are not as effective as this gold standard. Treatment using a pacemaker has been postulated as one of the potential methods. There were several hypotheses behind this theory. The surge of parasympathetic tone during sleep causes significant variation in heart rate. The hypothesis is that decreasing heart rate variation might cause decreased autonomic disturbance and improve respiratory function [12]. Second, AOP generates increased cardiac output. This can lead to a reduction in the chemoreceptor circulation time in the lungs. An increase in cardiac output accounts for a decrease in left ventricular filling pressure and hyperventilation. This combined effect on the lungs leads to a stabilizing respiratory system [38]. Our analysis supported the fact that pacing the atrium during the nighttime could reduce events of apnea and hypopnea.

We also analyzed the benefit of AOP depending on the severity of sleep apnea by using AHI = 30 as a cutoff between severe and non-severe sleep apnea syndrome. The result of this subgroup analysis pointed in a different direction compared with the primary analysis. AOP could not significantly reduce the respiratory event in each subgroup. This unexpected result from the subgroup analysis could be explained by three important studies (Guo et al. [32], Luthje et al. [22], Melzer et al. [23]) that were excluded from the subgroup analysis because the severity of sleep apnea syndrome was not reported. We might then imply that the severity of sleep apnea syndrome might not be the sole factor to play a role in sleep apnea.

The type of sleep apnea syndrome might affect the response to AOP in sleep disorder patients. At present, the benefits of treating central sleep apnea are still controversial. Modalities such as adaptive-servo ventilation and continuous positive pressure did not show significant cardiovascular benefit [39,40]. The successful treatment of CSA with cardiac resynchronization therapy in patients with low ejection fraction could therefore be explained by an improvement in heart failure symptoms, not the CSA itself [41,42]. The reason mentioned above might reduce the effectiveness of AOP. Future studies should compare the benefit of AOP in patients with OSA versus CSA.

Currently, due to the lack of evidence, there are no clinical practice guidelines recommending the use of AOP in patients with sleep apnea. Current practice defines optimal titration of treatment in patients, with a reduction in AHI < 5 after at least 15 min when treated with continuous positive airway pressure. In patients with severe symptoms (AHI > 30), reducing this index by half the baseline, or AHI < 10, is sufficient to define the response of treatment [43]. Although our meta-analysis showed a statistically significant reduction in AHI in patients treated with AOP, the effect of AOP was small, with SMD of AHI −0.29 (95%CI: −0.48 to −0.10, I^2^ = 57%). Therefore, AOP alone could not provide significant clinical benefits in treating patients with sleep apnea. To augment the effect of CPAP, AOP might be used as an adjunctive treatment in patients with sleep apnea who are already implanted with a permanent pacemaker.

There are several important limitations of our study. First, the size of each randomized study was small, with the largest study in our analysis containing only 20 patients. This could lead to a lack of statistical power for each study. Second, all the randomized studies had a crossover design with atrial pacing versus none for only a short period of time. Any long-term benefits are not possible to conclude from these studies. In addition, a wide variety of cardiac implantable electronic devices and different algorithms of atrial pacing were used in these studies. Third, the adverse effects of AOP were not mentioned in any report since AOP is believed to be a potential cause of atrial fibrillation [44]. Further details about pro-arrhythmia from AOP will have to be clarified. Fourth, the degree of heterogeneity across studies was statistically high (I^2^ = 57%). The possible source of this heterogeneity includes the difference in duration of treatment, which varied among studies ranging from 1 day to 210 days. Other sources of heterogeneity were differences in AOP protocol, type of pacemaker, and different types of sleep apnea syndrome. Lastly, not all studies were randomized controlled trials. There were two observational studies that were not able to support a causal relationship between AOP and reduction in respiratory events.

## 5. Conclusions

In summary, AOP was associated with a statistically significant reduction in AHI, but the magnitude of reduction was small. AOP alone would not provide significant clinical benefit in sleep apnea patients; however, the use of AOP might be beneficial as an adjunctive treatment in sleep apnea patients who were already implanted with a permanent pacemaker. Further knowledge about this issue needs to be clarified.

## Figures and Tables

**Figure 1 jcm-10-04065-f001:**
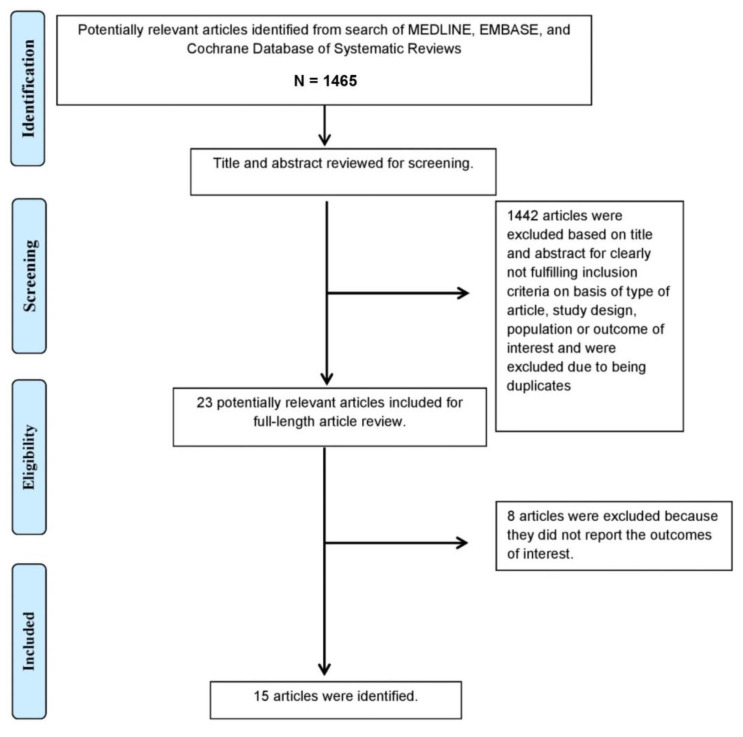
The literature retrieval, review, and selection process.

**Figure 2 jcm-10-04065-f002:**
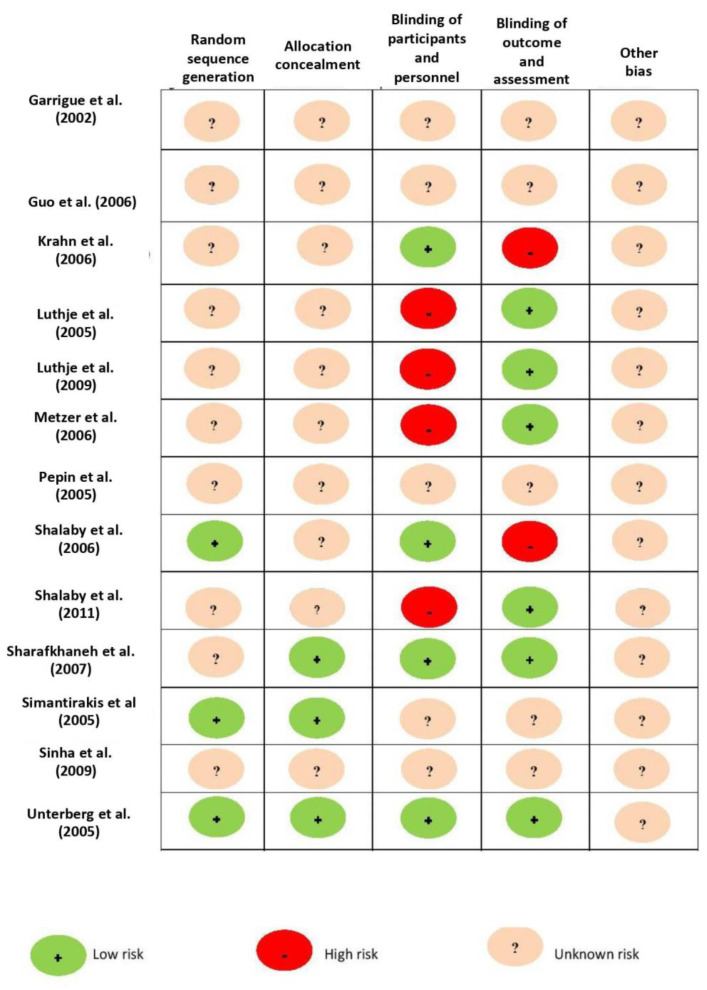
Cochrane risk of bias assessment for RCT. RCT: randomized controlled trial.

**Figure 3 jcm-10-04065-f003:**
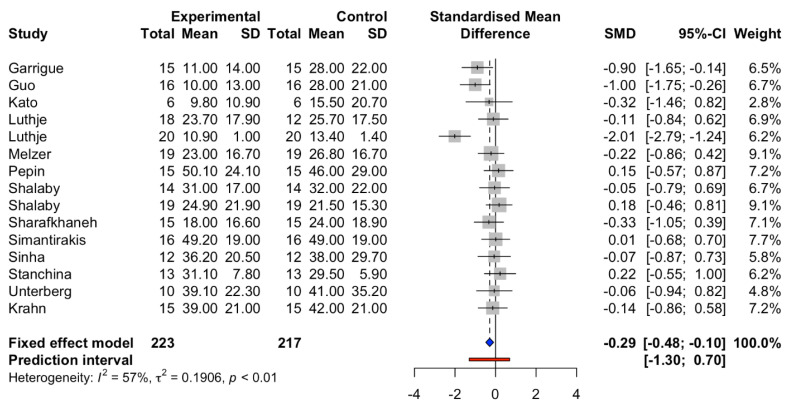
Standardized mean differences in AHI between atria overdrive pacing and control groups. Prediction interval is shown in red line. The blue square represents standard mean difference (SMD) and 95% Confidence interval (CI).

**Figure 4 jcm-10-04065-f004:**
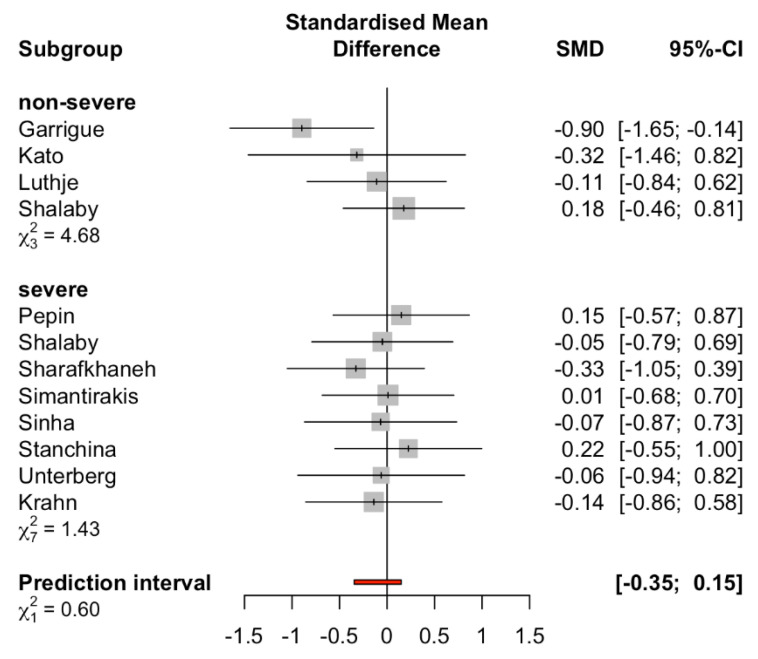
Subgroup analysis comparing severe sleep apnea syndrome and non-severe sleep apnea syndrome groups (cutoff AHI < 30). Prediction interval is shown in red line.

**Figure 5 jcm-10-04065-f005:**
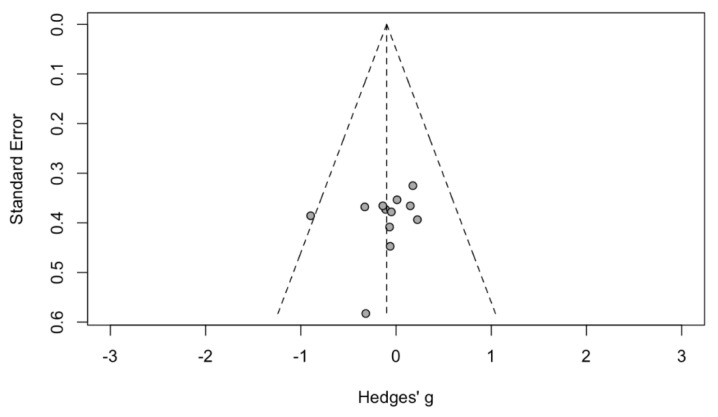
Funnel plot of AOP in sleep apnea patients. Circle showed published study.

**Table 1 jcm-10-04065-t001:** Characteristics of Meta-Analysis Studies. Characteristics of Included Studies.

	Garrigue et al., (2002)	Guo et al., (2006)	Kato et al., (2001)	Krahn et al., (2006)	Luthje et al., (2005)	Luthje et al., (2009)	Metzer et al., (2006)	Pepin et al., (2005)	Shalaby et al., (2006)	Shalaby et al., (2011)	Sharafkhaneh et al., (2007)	Simantirakis et al., (2005)	Sinha et al., (2009)	Stanchina et al., (2007)	Unterberg et al., (2005)
**Country (year)**	France 2002	China 2006	Japan 2001	USA Canada UK 2006	Germany	Germany	Germany	France	USA	USA	USA	Greece	Germany	USA	Germany
**Type of sleep apnea**	OSA and CSA	OSA and CSA	OSA and CSA	OSA	OSA and CSA	CSA	OSA and CSA	OSA	OSA	OSA and CSA	OSA	OSA	OSA and Mixed type	OSA	OSA
**Sleep apnea severity**	AHI baseline 9 ± 4	NR	NR	AHI 34 ± 14	NR	AHI 26 ± 18.2	NR	46.3 ± 28.5	35.2 ± 21.9	AHI 21.5 ± 15.3	AHI 34.8 ± 15.5	AHI 49 ± 19	AHI 38 ± 29.7	AHI 40.9 ± 6.4	AHI 41 ± 16
**Age**	69 ± 9	NR	74 ± 2	60 ± 13	63.2 ± 1.7	66.1 ± 9.8	68 ± 11.4	71 ± 10	66 ± 12	67.2 ± 7.5	74 ± 6.6	60 ± 11	61 ± 10	68.6 ± 3.7	61 ± 5.6
**BMI**	NR	NR	24.4 ± 2.7	29.9 (21.7–42.0)	30.1 ± 0.8	27.1 ± 4.2	29.5 ± 6	27 ± 3	32 ± 2	28.1 ± 3.5	29 ± 4.4	NR	28.9 ± 6.5	28.7 ± 1.5	33 ± 5.44
**Intervention/** **control**	AOP/No pacing TypeP	AOP/No pacing Type P	Physiologic pacing increase HR/No pacing Type P	AOP/No pacing Type P	AOP/No pacing Type P	CRT+AOP/CRT	AOP/No pacing Type P	AOP/No pacing Type P	AOP/No pacing Type P	CRT+AOP/CRT	AOP/No AOP Type P	AOP/No AOP Type P	AOP/No AOP Type P	CRT + AOP/CRT	AOP/No AOP Type T
**Treatment time**	3 days	3 days	1 week	2 days	3 days	2 days	7 days	30 days	1 day	84 days	3 days	60 days	210 days	180 days	3 days
**Cases (n)**	15	16	6	15	20	30	19	15	14	19	15	16	12	13	10
**Control (n)**	15	16	6	15	20	30	19	15	14	19	15	16	12	13	10
**Mean difference (*p* value)**	*p* < 0.01	*p* < 0.01	*p* < 0.05	*p* = 0.23	*p* < 0.01	*p* < 0.01	*p* = 0.49	NS	*p* = 0.8	*p* = 0.45	*p* < 0.05	*p* = 0.87	NS	*p* = 0.02	*p* = 0.002
**AHI post treatment**	Total AHI: 28 ± 22 (No pacing) versus 11 ± 14 (AOP)	Total AHI 28 ± 21 (No pacing) 10 ± 13 (AOP)	Total AHI 15.5 ± 20.7 (No pacing) 9.8 ± 10.9 (pacemaker increased heart rate)	Total AHI 38.6 ± 20.5 (pacing 75 bpm) versus 42.1 ± 20.7 (pacing off), MD −3.4, 95% CI: −9.3 to 2.5,	Total AHI 20.9 ± 2.1 (No pacing) versus 19.5 ± 2.4 (AOP 7 bpm) versus 17.8 ± 1.9 (AOP 15 bpm), NS	Total AHI: 37.1 ± 13.4 (CRT) versus 25.7 ± 17.5 (CRT + AOP)	Total AHI 26.8 ± 17.1 (control) versus 23 ± 16.7 (AOP)	Total AHI 46 ± 29 (control) versus 50 ± 24 (AOP)	Total AHI 32 ± 22 (pacing) versus 34 ± 22 (AOP-10 bpm) versus 31 ± 17 (AOP-20 bpm)	Total AHI 21.5 ± 15.3 (CRT) versus 24.9 ± 21.9 (AOP + CRT)	Total AHI 24 ± 19.8 (Control) 18 ± 16.6 (Pacing)	Total AHI 49.2 ± 19 (Pacing) versus 49 ± 19 (No pacing)	Total AHI: 38 ± 29.7 (Control) 26.2 ± 20.5 (Pacing)	Total AHI 29.5 ± 5.9 (Control) 31.1 ± 7.8 (Pacing)	Total AHI 41 ± 35.2 (Control) 39.1 ± 22.3 (Pacing)
**Type of study**	RCT, crossover	RCT, crossover	Observational	RCT, crossover	RCT, crossover	RCT, crossover	RCT, crossover	RCT, crossover	RCT, crossover	RCT, crossover	RCT, crossover	RCT, crossover	RCT, crossover	Observational	RCT, crossover

Abbreviations: AOP: atrial overdrive pacing, AHI: apnea–hypopnea index, BMI: body mass index, CRT: cardiac resynchronization therapy, CSA: central sleep apnea, P: permanent pacemaker, T: temporary pacemaker, MD: mean difference, NR: not reported, NS: not significant, OSA: obstructive sleep apnea, R: randomized trial, RCT: randomized controlled trial.

## Data Availability

The data for this systematic review and all potentially eligible studies are publicly available through the Open Science Framework (URL: osf.io/3f2pb, accessed on 27 August 2021).

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
