# Peer review of "Benefit of Atrial Overdrive Pacing in Patients with Sleep Apnea: A Meta-Analysis"

_jcm, 2021, doi:10.3390/jcm10184065_

Round 1

Reviewer 1 Report

The authors present an interesting meta-analysis about the potential of atrial overdrive pacing in patients suffering from sleep apnea. Although the authors found a significant treatment effect, the effect itself was rather small. Hence, the clinical implication is rather limited, which should also be discussed more clearly. Generally, the methodology seems appropriate, but some major aspects remain to get addressed:

  1. “Sleep apnea is divided 41 into 2 subtypes, obstructive sleep apnea (OSA) and central sleep apnea (CSA)“ - Sleep apnea is categorized into 3 subgroups: central, obstructive and mixed sleep apnea. Please adapt accordingly.
  2. “The total 47 number is varying from 711 million and 961 million in every country” - This sentence is unclear to understand. Please clarify.
  3. "From current knowledge, early diagnosis and treatment of patients with sleep apnea would gain significant benefits both medically and economically." Please provide a reference, otherwise adapt the phrasing.
  4. Proper description of AOP in your introduction is missing completely. You should not just mention how it is delivered, but why it was developed and what it should achieve.
  5. Please separate central and obstructive sleep apnea more clearly throughout your manuscript. I also strongly recommend you to discuss this more in detail, as this would clearly enhance the value of your manuscript. Up to my knowledge, there is no therapy yet to target isolated central sleep apnea, which has proven benefits in hard endpoints. Most trials have shown a benefit for CSA due to treating HF. This was for example shown in de novo CRT (Europace 2011; 13(8):1174-9) and upgrading to CRT (Am J Cardiol. 2021; 139:97-104.). Otherwise, adaptive servoventilation (N Engl J Med. 2015; 373:1095-105) and several other treatment modalities including transvenous phrenic nerve stimulation (meta-analysis: Am J Cardiol. 2020; 127:73-83) failed to improve cardiovascular mortality.
  6. Provide a rationale why different searching criteria were used for literature research in Cochrane library?
  7. Provide a rationale for rather preferring a random effect model over a fixed one?
  8. As already stated above it is rather keen to start your discussion stating that AOP has shown a benefit for patients suffering from sleep apnea given a mean difference of around -0.29 in AHI. Therefore, I recommend softening all of these statements and always put them put into context of other available therapies (CPAP, HF therapy, etc.). Both trials described above have shown either a median reduction of 16.9 in CRT upgrades or a mean reduction of 13.1 in de novo implantations. Please also include mean/median differences in AHI due to CPAP in OSA reported by recent trials.
  9. "The most effective method in treating patients with sleep apnea is continuous positive pressure (CPAP)." – Again, please always separate OSA and CSA as they offer different options of treatment, especially in the setting of HF.
  10. Review of language is advised.

Reviewer 2 Report

Manuscript entitled „Benefit of Atrial Overdrive Pacing in Patients with Sleep Apnea: A Meta-Analysis” reports on the impact of atrial overdrive pacing among patients with sleep apnea.

The manuscript is not properly structured. There is not enough background provided in the beginning to allow proper understanding of the topic. Cardiovascular OSA comorbidities needing AOP should be explained and mechanisms responsible should be also included (fe. doi: 10.3389/fneur.2018.00635).

In abstract, when defining OSA severity it should be referred to AHI, not only sated a sole number.

Throughout the manuscript the abbreviations are either not introduced and used or are introduced and later the full form is used, this should be corrected.

Round 2

Reviewer 1 Report

Thank you for revising your manuscript. Most aspects raised were addressed appropriately, besides a final English language editing I have a single remaining comment:

- Table 1: As you have inserted the types of sleep apnea included in each single study, you should use the term "sleep apnea severity" instead of "OSA severity" in the row below.

Author Response

- Table 1: As you have inserted the types of sleep apnea included in each single study, you should use the term "sleep apnea severity" instead of "OSA severity" in the row below.

Response: We have corrected the table 1 changing from OSA severity to Sleep apnea severity.

We have comprehensively reviewed and made additional corrections as requested including language editing.